# Identification of the Actin-Binding Region and Binding to Host Plant Apple Actin of Immunodominant Transmembrane Protein of ‘*Candidatus* Phytoplasma mali’

**DOI:** 10.3390/ijms24020968

**Published:** 2023-01-04

**Authors:** Kajohn Boonrod, Linda Kuaguim, Mario Braun, Christine Müller-Renno, Christiane Ziegler, Gabi Krczal

**Affiliations:** 1RLP-AgroScience GmbH, AlPlanta-Institute for Plant Research, Breitenweg 71, 67435 Neustadt, Germany; 2Department of Physics and Research Center OPTIMAS, University of Kaiserslautern, Erwin-Schrödinger Str. 56, 67663 Kaiserslautern, Germany

**Keywords:** phytoplasma, immunodominant membrane protein, actin, single-molecule force spectroscopy

## Abstract

‘*Candidatus* Phytoplasma mali’ (‘*Ca*. P. mali’) has only one major membrane protein, the immunodominant membrane protein (Imp), which is regarded as being close to the ancestor of all phytoplasma immunodominant membrane proteins. Imp binds to actin and possibly facilitates its movement in the plant or insect host cells. However, protein sequences of Imp are quite diverse among phytoplasma species, thus resulting in difficulties in identifying conserved domains across species. In this work, we compare Imp protein sequences of ‘*Ca*. P. mali’ strain PM19 (Imp-PM19) with Imp of different strains of ‘*Ca*. P. mali’ and identify its actin-binding domain. Moreover, we show that Imp binds to the actin of apple (*Malus x domestica*), which is the host plant of ‘*Ca*. P. mali’. Using molecular and scanning force spectroscopy analysis, we find that the actin-binding domain of Imp-PM19 contains a highly positively charged amino acid cluster. Our result could allow investigating a possible correlation between Imp variants and the infectivity of the corresponding ‘*Ca*. P. mali’ isolates.

## 1. Introduction

Phytoplasmas are plant pathogenic Gram-positive eubacteria belonging to the class *Mollicutes* and were assigned to the novel provisional genus ‘*Candidatus* Phytoplasma’ [1,2]. Phytoplasmas reside in the phloem of infected plants and cause various diseases in several hundreds of plant species [3]. They are transmitted by phloem-feeding hemipterous insects, mainly leafhoppers (*Cicadellidae*), planthoppers (*Fulgoroidea*), and psyllids (*Psyllidae*).

Phytoplasmas cause yield losses in crops, such as apple trees, worldwide by affecting fruit, flower, and seed formation [3]. Apple proliferation is one of the most important apple diseases caused by ‘*Ca*. P. mali’ and affects almost all apple cultivars. The infected plants show witches’ broom disease (increasing number of shoots with bushy, dwarfed, branching features) and enlarged stipule. The fruits of infected plants decrease in size (up to 50%), weight (up to 70%), and fruit quality, thus leading to massive yield losses and economic damage in apple production [4].

All phytoplasmas lack a cell wall, and their single-cell membrane is in direct contact with the host environment. Thus, phytoplasma membrane proteins play a major role in phytoplasma–host interactions, but little is known about their multiple functions. Serological studies revealed that immunodominant proteins (IDP) represent a major portion of the total cellular membrane proteins in most phytoplasmas, and genes encoding IDPs were isolated from the members of several phytoplasma groups [5]. The corresponding sequence alignment data indicate that they are not orthologues of each other and that nonhomologous proteins play the role of IDP in various phytoplasmas. These IDPs differ enormously in amino acid (aa) composition and antigenic variations. However, they were classified into three different groups: (1) Imp (immunodominant membrane protein), (2) IdpA (Immunodominant membrane protein A), and (3) Amp (antigenic membrane protein), reviewed by [6,7]. Although IDPs are variable in protein sequences, they share the same general organization composed of a central hydrophilic region, which may reside on the outside of the phytoplasma cell, and one or two transmembrane regions [8]. While homologous genes of Imp have also been detected in IdpA encoding phytoplasmas, no orthologous genes to either IdpA or Amp have been identified in the genome sequence of ‘*Ca.* P. mali’ [6,9]. Therefore, it has been suggested that Imp represents an ancestral type of phytoplasma IDPs [10].

Interestingly, Amps of onion yellows phytoplasma (OY) and ‘*Ca*. P. asteris’ interact with actin of their leafhopper vector species [11,12]. These results suggest that an interaction between the phytoplasma Amp and the host’s actin or microfilament could be essential for phytoplasma transmission by insect vectors [12] and crucial for the intracellular motility of phytoplasmas in the insect vector [11]. It was reported that Imp and Amp were also co-precipitated with other insect host proteins [12,13,14] and that Amp of ‘*Ca.* P. asteris’ specifically binds to the α and ß subunits of ATP synthase of its leafhopper vector species [11].

It was shown that Imp-PM19 binds to the actin filament of *N. benthamiana* plants and the transgenic plants expressing the protein did not show any disease like symptoms [15]. Thus, it was speculated that the Imp-PM19-actin-binding could also play a role in phytoplasma motility in plants, rather than as a pathogenic effector [15].

In the present study, we examined the functional properties of Imp in more detail. Since the location of a putative actin-binding region within the Imp-PM19 protein is unknown, we aligned the Imp protein sequences of different ‘*Ca*. P. mali’ strains and identified an actin-binding region in vivo and in vitro by confocal microscopy and a co-sedimentation assay. Additionally, we applied single-molecule force spectroscopy (SMFS, see, e.g., [16]) to the most promising Imp mutant (missing actin-binding site) compared to the Imp wildtype to show the reduced interaction.

## 2. Results

### 2.1. Amino Sequence Alignment

To identify the actin-binding region of Imp-PM19, the amino acid sequences of Imp-PM19 derived from different ‘*Ca.* P. mali’ strains were aligned. The result in Figure 1 shows high variability in the Imp-PM19 amino acid sequences among ‘*Ca.* P. mali’ strains. However, we can cluster three conserved regions of Imp-PM19 (Figure 1a,b). We further aligned the aa sequence of Imp-PM19 against Amp-OY which is known to have an actin-binding region. As we expected, we found that none of the aa regions of Imp-PM19 is homologous with the Amp actin-binding region.

### 2.2. Actin-Binding Activity of Imp-PM19 in Planta

To identify an actin-binding region of Imp-PM19, we generated *Imp-PM19* mutants (ΔD1-∆D3), as shown in Figure 1c. The aa of conserved regions were randomly mutated to change the total net charge of the regions. The *Imp-PM19* and *ΔD1-∆D3*-*Imp-PM19*-mutants were transiently expressed as GFP fusion proteins in *N. benthamiana* plants via agrobacterium infiltration. The expression and localization of Imp-PM19 and the Imp-PM19 mutants were visualized with a confocal microscope by comparison with the localization of an actin-binding protein (mTalin) fused with red fluorescence protein (RFP), mTalin-RFP. The result of the localization of Imp-PM19 indicated that Imp-PM19 and ΔD1 and ΔD2-Imp-PM19 mutants (Figure 2) bound to *N. benthamiana* actin, while the ∆D3-Imp-PM19-mutant was dispersed within the cytoplasm. This suggests that the amino acid cluster is located in the C-terminal part of Imp-PM19 (D3) is the actin-binding region.

### 2.3. Expression and Purification of the Recombinant Imp-PM19 and ∆D3-Imp-PM19 Mutant Proteins as His-Tag Fused Protein

We further confirmed the in vivo result of Imp-PM19-actin binding region in vitro by performing a co-sediment assay as described by Boonrod et al. [14]. The *Imp-PM19* and *∆D3-Imp-PM19* mutant were recombinantly expressed in *E. coli* as Hexa-histidine (6XHis)-tag fused proteins for purification. The result in Figure 3 shows that the proteins were expressed and purified as native soluble proteins.

### 2.4. Co-Sediment Actin Binding Assay

Since apple is the natural host of ‘*Ca.* P. mali’, we intended to demonstrate that Imp-PM19 could also bind actin of apple plant (actin_apple_). Thus, we first recombinantly expressed actin_apple_. Hatano and co-workers [17] demonstrated that human actin could be functionally expressed using the *P. pastoris* expression system, thus we adopted this expression system to produce an octa-histidine (8XHis) tagged actin_apple_. After a protease treatment to remove the His-tag, coomassie brilliant blue staining and Western blot analysis showed that the actin_apple_ was successfully purified (Figure 4a) and could form F- actin in vitro (Figure 4b). The F-actins were further confirmed by staining with phalloidin, Acti-stain™ fluorescent phalloidin 555 (Figure 4c). The rabbit muscle F actin (F-actin_rabbit_, a positive control) and the recombinant actin_apple_ were then used in a co-sediment assay. The result shows that Imp-PM19 bind to both F-actins, while the ∆D3-Imp-PM19-mutant lost its actin-binding activity (Figure 4c). Thus, this result confirms the results of the Imp-PM19-actin-binding assay in vivo (Figure 2).

### 2.5. Measuring of Interaction Forces between Imp-PM19, ∆D3-Imp-PM19 Mutant and F-Actin Filament Protein by Scanning Force Microscopy (SFM)

Single-molecule force spectroscopy (SMFS) measurement is used to detect and measure the interaction force between one ligand molecule on the tip and one receptor protein on the surface. To achieve this goal, two important conditions must be fulfilled. First, the ligand must be able to move freely to connect the receptor with the tip. Second, the surface of the substrate must be fully recovered with the receptor protein to avoid the interaction of the ligand with the naked substrate. The latter condition requires a high amount of actin. The result (Figure 4) shows that Imp-PM19 binds actin_rabbit_ not significantly different to actin_apple_. Moreover, the two actin sequences recognized by Imp-PM19 are highly conserved (Figure 4e). Thus, we used actin_rabbit_ instead of actin_apple_ for SFM analysis_._

In the first step, a short scan of the surface was performed with naked functionalized tips to identify actin filaments. Subsequently, force curves were taken at a position on the filament (not shown). Comparing the obtained measurements and force–distance curves on a pure poly-l-lysine surface as a negative control (higher tip adhesion between 300–400 pN, observed stiff jump and small interactions on the extend curve) and the F-actin_rabbit_ immobilized surface, the typical shape of curves obtained with the actin filament could be distinguished from those obtained on a poly-l-lysine surface (negative control). Thus, it was confirmed that the functionalized cantilever tips could detect the actin filament on a mica plate. To measure the binding of Imp-PM19 respectively ∆D3-Imp-PM19 and actin filament, the Imp-PM19 and ∆D3-Imp-PM19 mutant were covalently bound to the functionalized SFM tip. The result (Figure 5) shows that a thousand single-molecule forces curves could be measured on the surface of F-actin_rabbit_ with a functionalized Imp-PM19-SFM-tip (Curves without events/curves with the tip adhesion were removed during the data analysis). Then, the unbinding events between Imp-PM19 and ∆D3-Imp-PM19 mutant-F-actin_rabbit_ were detected after 10 nm. The force-distance curves obtained with the Imp-PM19 (Figure 5a) show a retract curve with many unbinding steps compared to those obtained with the ∆D3-Imp-PM19 mutant (Figure 5b). Imp-PM19 in contact with F-actin_rabbit_ filament shows significant rupture events with an unbinding length between 10 and 120 nm, and a minor interaction is observed occasionally between 520 and 540 nm. In addition, the dissociation of the Imp-PM19-F-actin_rabbit_ filament complex shows a rupture force of 100 pN at a tip-mica distance of 100 nm. The unbinding force of 100 pN between Imp-PM19 and F-actin protein is similar to the reported values in the case of single receptor-ligand pairs via a heterobifunctional polyethylene glycol (PEG) derivative [18] and antibody and antigen (anti-actin and actin, [19]). In addition, two exponential changes in the slope are observed by the Imp-PM19 force-distance curve at ~50 and 100 nm (before the rupture force) during the retraction process before the rupture force.

In contrast, the ∆D3-Imp-PM19 mutant shows in contact with the F-actin_rabbit_ filament an unbinding length of 15 nm (10–25 nm). The absence of rupture events of the ∆D3-Imp-PM19 mutant suggests that as soon as the ∆D3-Imp-PM19 mutant functionalized SFM tip meets the F-actin_rabbit_ filament, it leaves directly without interaction.

We further investigated other parameters, such as most frequently observed adhesion force and work of adhesion, to confirm the interaction of the Imp-PM19 with the F-actin_rabbit_ filament.

The results in Figure 6 show that the interaction force of the Imp-PM19 with the F-actin_rabbit_ filament occurs between 40 and 400 pN, with a most frequently observed adhesion around 80–90 pN, whereas the ∆D3-Imp-PM19 mutant interaction is observed between 0–60 pN, with a most frequently observed adhesion force around 0–10 pN. In addition, the maximal work of adhesion by Imp-PM19 occurs between 4.5 and 5 aJ and by ∆D3-Imp-PM19 mutant at 1–2 aJ, suggesting that more work is needed to separate the Imp-PM19 from the Imp-PM19-F-actin_rabbit_ filament complex. “Higher values of the work of adhesion” could also mean that Imp-PM19 is specifically tied to the F-actin_rabbit_ filament or maybe the Imp-PM19 interacts more with the F-actin_rabbit_ filament than the ∆D3-Imp-PM19 mutant. Thus, the results of SMFS analysis strongly support the results of the molecular analysis.

### 2.6. Imp Protein Folding Predictation

Although crystallo-graphic analysis of Imp has not yet been investigated, the use of a reliable computer program for protein folding analysis such as AlphaFold or Phyre2 could provide an overview of how the protein might fold. In order to compare the protein folding of Imp-PM19 and the ∆D3-Imp-PM19 mutant, we used the computer program Phyre2 [20] to predict protein folding. The result in Figure 7 shows that Imp can fold as an alpha-alpha superhelix. However, mutation of the actin binding region in the ∆D3-imp PM19 mutant did not alter the predicted protein folding. Thus, the loss of actin binding ability in the ∆D3-Imp-PM19 mutant may not be due to altered protein folding.

## 3. Discussion

IDPs play an important role in the phytoplasma–host interaction. More specifically they seem to be involved in the attachment of the phytoplasmas to the host cell surface via actin (reviewed by [7]). Thus, the actin-binding of IDPs could be a general feature for phytoplasma motility in plants and insects.

In the present work, we aligned the protein sequences of Imp-PM19 against Imp of different ‘*Ca.* P. mali’ strains. We show that besides the TM domain, which is highly conserved, there are 3 different conserved regions (ΔD1-∆D3). Using the transient expression of Imp-PM19 and the Imp-PM19 mutants fused GFP in planta, the result clearly indicates that the D3 region of Imp-PM19, comprising amino acid 135–162, is an actin-binding region, as further confirmed by a co-sediment assay with apple and rabbit muscle F-actin. Several actin-binding regions were already identified from different actin-binding proteins (reviewed by Maciver and co-workers) [21]. However, the general features of actin-binding regions vary among the actin-binding proteins. In several classes of actin-binding proteins, the sequence of the actin-binding regions comprises 10 and 30 residues [22]. D3 of Imp-PM19 comprises 26 aa residues comprising several lysine and charged amino acid residues. Modifying this region into a region of low charge in the ∆D3-Imp-PM19 mutant abolished the actin-binding activity of Imp-PM19 in in vivo and in vitro assays. The SMFS analysis results confirm that Imp-PM19 binds to the filament actin, and their dissociation shows rising rupture events with an unbinding or ruptures force of 100 pN. This unbinding force is similar to the reported values in the case of single receptor-ligand pairs via a heterobifunctional polyethylene glycol (PEG) derivative [18] and antibody and antigen (anti-actin and actin, [19]). On the other hand, the ∆D3-Imp-PM19 mutant shows a most frequently observed adhesion force in the distance interval of its unbinding length. The more significant length of unbinding by Imp-PM19 and the F-actin_rabbit_ filament can emphasize that Imp-PM19 and F-actin are bound together, and their dissociation leads to the breaking of the binding through stretching of Imp-PM19 to the F-actin_rabbit_ filament over a distance. The stretching (spacer and the complex Imp-PM19-F-actin) can also be explained by the observed exponential change in the slope by Imp-PM19 force-distance curve in the retraction process. This change can result from a decreased effective spring constant and is a feature of the specific unbinding force [23]. In contrast, the force-distance curve observed with ∆D3-Imp-PM19 mutant underlines the character of a non-specific interaction or unbinding force because the same slope is kept during the retraction process.

Therefore, the higher value of the work of adhesion observed by Imp-PM19 confirms that more work is needed to separate Imp-PM19 to its cognate F-actin_rabbit_ filament if compared to the mutant. A value of *p* ≤ 0.05 (alpha level) was considered statistically significant. All results together strongly indicate that D3 of Imp-PM19, which contains charged amino acid residues, is involved in the actin binding of Imp-PM19. The same actin-binding feature was also reported for villin, an actin-binding protein of the intestinal brush border [24]. Moreover, our results show that beside actin_rabbit_, Imp-PM19 can bind to plant actins, including *N. benthamiana* actin in the in vivo assay and actin_apple_ in the co-sediment assay. This suggests that Imp-PM19 can bind actin across kingdoms. Based on these data, it can be postulated that Imp-PM19 might also bind insect actin as described for Amp of onion yellows phytoplasma (OY) and Ca. P. asteris [12,13]. However, this must be experimentally confirmed. In *spiroplasma citri*, the minimal actin-binding region of Phosphoglycerate Kinase (PGK) is involved in the transmission of this phytoplasma by its leaf hopper vector [25]. Therefore, it cannot be ruled out that the actin-binding region of Imp-PM19 of ‘*Ca*. P. mali’ could play a role in its transmission by its insect vectors. Due to lacking genomic data on *C. picta* and *C. melanoneura*, the insect vectors of ‘*Ca*. P. mali’, it is currently difficult to isolate their actin genes for recombinant expression in further studies.

Some intracellular bacteria stimulate actin to drive the rocketing motility by its polymerization [26,27]. In plants, inhibiting actin polymerization or actin silencing [28] reduces the movement protein (MP) virus particle trafficking of *Tobacco mosaic virus* [29]. Amp binds actin of insects, which are only in OY-transmitting leafhopper species [11,12]. This binding was not reported to exhibit any negative effect on the life cycle of the host insect. Thus, these results suggested that the actin-binding activity of Amp could be essential for phytoplasma transmission by insect vectors [12]. Phytoplasmas lack genes coding for movement. Actin-binding feature could support the movement of the phytoplasmas in the phloem, and thus the colonization of the plant host.

In conclusion, the actin binding region of Imp-PM19 is located at C-termini of the protein. The identified actin binding region was confirmed in vitro and in vivo. Imp-PM19 binds not only rabbit muscle actin, but also apple actin, a host plant, thus its binding to actin of transmission insects could be postulated. The identification of an actin-binding region of Imp-PM19 of ‘*Ca.* P. mali’ will help to increase the knowledge of phytoplasma-host interactions. Moreover, our results can be used to select a highly specific molecule (single chain antibody, nobody, peptide) that binds to the identified actin-binding region to develop resistance strategies to combat phytoplasma infection. Moreover, the gained information may be used to examine a possible link in the role of the Imp-PM19 protein of phytoplasma strains in pathological traits and transmission.

## 4. Materials and Methods

### 4.1. Isolation of the Imp-PM19 Gene of ‘Ca. P. mali’

The Imp-PM19 gene was isolated from a ‘*Ca.* P. mali’ strain PM19 which was recently transmitted by field-collected overwintered adults of *Cacopsylla picta* (*C. picta*) to healthy test plants of *Malus x domestica* cv. Golden Delicious [30]. Total DNA was extracted from phloem preparations of ‘*Ca.* P. mali’ strain PM19-infected plants with a CTAB-based protocol [30]. The Imp-PM19 gene was amplified from the total DNA extract with primers f318A/r318B, as reported by [31]. The cycling conditions were modified to 1 min 95 °C followed by 40 cycles of 15 s 95 °C, 20 s 52 °C, and 1 min 69 °C and a final step of 4 min 72 °C. PCR products were cloned into the pGEMT-easy vector according to the manufacturer’s instruction (Promega), yielding pGEMT-Imp-PM19. Three independent clones were Sanger sequenced (4base lab AG, Reutlingen, Germany, http://www.4base-lab.de) with standard procedures in both directions.

### 4.2. Imp-PM19 Protein Alignment

Imp protein sequences of 5 different ‘*Ca*. P. mali’ strains (Accession number; AXY94996, CBJ17008, AXY94998, WP0125024208, CB170454) were aligned against the Imp-PM19 protein sequence using the DNA star program. All Imp protein sequences were obtained from the NCBI database.

### 4.3. Cloning of Imp-PM19 and Mutants in Expression Vectors

The *Imp-PM19* gene was amplified from the pGEMT-Imp-PM19 plasmid by PCR with primers introducing restriction sites at the 5′ and 3′ ends of the *Imp-PM19* coding sequence (5′-BamHI and 3′-XhoI sites for cloning into expression vector pET23a+). The PCR products were directly cloned into a pJet PCR cloning vector (Thermo Fisher Scientific, Karlsruhe, Germany), resulting in a pJet-BamHI-*Imp-PM19*-XhoI. In order to fuse *Imp-PM19* with hexa-histidine (6XHis), the *Imp-PM19* gene was sub-cloned into pET23a+ (Novagen, Merck KGaA, Darmstadt, Germany) at corresponding restriction sites, resulting in pET23a+: *Imp-PM19-His*. The ∆*D1-*∆*D3-Imp-PM19* mutants were designed, and the gene was synthesized by Gene Script (USA). The genes were sub-cloned in expression vectors as described for Imp-PM19.

### 4.4. Protein Expression and Extraction

For protein expression, the *E. coli* strain BL21 (DE3) was used. Cells were grown in LB media containing 100 µg/mL ampicillin. At an OD_600_ = 0.6, protein expression was induced by adding isopropyl-β-D-thiogalactopyranoside (IPTG) to a final concentration of 1 mM. After incubation at 37 °C for 2 h or at 14 °C overnight, cells were harvested and resuspended in BugBuster^TM^ Protein Extraction Reagent (Novagen, Merck KGaA, Darmstadt, Germany, 10 mL/g of cells) containing 5 µL Benzonase (25 u/µL, (Novagen, Merck KGaA, Darmstadt, Germany)), 10 mM DTT, and one tablet of complete protease inhibitor (EDTA-free, Roche, Grenzach-Wyhlen, Germany). The resuspended cells were incubated for 1 h at 4 °C under agitation. The lysate was centrifuged at 9000× *g* for 10 min and the soluble fraction was purified using amylose magnetic beads (New England Biolabs, Frankfurt am Main, Germany) or Ni-NTA agarose (Qiagen, Hilden, Germany) and further analyzed by SDS-PAGE.

### 4.5. Purification of Bacterial Expressed Imp-PM19

The bacterial expressed Imp-PM19 and ∆D3 Imp-PM19 mutant were purified using Ni-NTA as described by Boonrod and co-workers [14]. The eluted fusion protein was analyzed by SDS-PAGE.

### 4.6. Cloning Gene Encoded Actin_apple_ into P. Pastoris Expression Vector

The expression and purification actin were performed as described by Hatano and co-workers [17]. The pPICZ plasmid used in his work, pPICZc-*HsACTB*-thymosinB-8XHis* (Addgene, Watertown MA, USA). For the replacement of human actin B in pPICZc-*HsACTB*-thymosinB-8XHis*, a gene *Malus x domestica* actin 1 (*actin_apple_*_,_ Accession Number XM_008356922) was codon optimized for *P. pastoris* expression and synthesized (GeneArt™ Strings™ service by Thermo Fisher Scientific, Karlsruhe, Germany) with the addition of the restriction sites EcoRI and NdeI at the 5′ and 3′ end, respectively. The digested EcoRI-NdeI actin gene fragments were then sub-cloned into the pPICZc-*HsACTB*-thymosinB-8XHis* destination vector, resulting in pPICZc-*actin_apple_-thymosinB-8xHis*. The plasmids were sequenced using specific alcohol oxidase 1 (AOX) promoter (5′-GACTGGTTCCAATTGACAAGC and terminator primers 3′-GCAAATGGCATTCTGACATCC).

### 4.7. Transformation of Pichia Pastoris and Protein Expression and Purification

#### 4.7.1. Transformation

Competent *Pichia pastoris* (*P. pastoris*) cells were prepared and transformed using the Pichia EasyComp™ kit (Thermo Fisher Scientific, Karlsruhe, Germany), following the provided manual and protocols. The pPICZα vector carrying the actin genes was linearized using the BstXI restriction enzyme prior to transformation according to the manual for the pPICZα A, B, and C expression vectors (Thermo Fisher Scientific, Karlsruhe, Germany). Zeocin™ resistant clones were screened for the presence of actin by Colony-PCR. The positive clones were used for actin expression analysis in small-scale (2 mL) cultures.

#### 4.7.2. Expression and Purification of Recombinant Actin from Pichia Pastoris

A 25 mL pre-culture in YPD liquid medium containing 100 mM Zeocin™ (YPD-Zeo100) was inoculated from the glycerol stock and incubated for 2 days shaking with 180 RPM at 30 °C. 4 mL of this pre-culture was then used to inoculate 400 mL YPD-Zeo100 medium (1:100), which was split into 2 baffled 500 mL culture flasks for better aeration, with 100µL of paraffin oil added to each flask as an antifoaming agent. These cultures were then incubated for 2 days, shaking with 180 RPM at 30 °C to build biomass. The cells were then pelleted for 10 min at 11,000× *g* at room temperature and washed once with distilled water before resuspension in fresh YPD-Zeo100 supplemented with 0.5% HPLC grade methanol for induction of the AOX promoter and paraffin oil was added as before. Expression was carried out for 2 days with daily addition of 0.5% methanol to compensate for metabolism and evaporation. The culture was centrifuged again, and the resulting pellet was weighed. We added 3 g of 250 µm glass beads along with 10 mL extraction buffer (50 mM sodium phosphate pH 8.0, 300 mM sodium chloride, 10 mM imidazole, 1 mM phenylmethylsulfonyl fluoride (PMSF), 0.8 mM adenosine triphosphate (ATP), 0.5 mM dithiothreitol (DTT), and Complete™ protease inhibitor cocktail) per gram of wet weight. Cell lysis was performed for 5 min in a bead beater homogenizer. The glass beads were removed by filtration through a fleece fitted funnel, and the flow through was centrifuged at 11,000× *g* for 10 min at 4 °C to remove most of the debris. The supernatant was then further cleared by centrifugation at 43,000× *g* for 30 min at 4 °C in a Sorvall RC 5C plus centrifuge using an SS-40 rotor. The ultra-cleared supernatant was carefully transferred to a fresh tube, 1 mL Ni-NTA resin (Qiagen, Hilden, Germany) was added and shaken gently for 1 h at 4 °C. The resin was pelleted by centrifugation at 1260× *g* for 5 min at 4 °C, resuspended in 10 mL ice-cold extraction buffer and loaded onto a small glass chromatography column. The buffer was drained, and the packed resin column was washed with 20 mL ice-cold buffer G (5 mM Tris-HCl pH 8.0, 0.2 mM CaCl2, 0.5 mM DTT, 0.2 mM ATP and 0.01% NaN_3_). After draining, the column was closed with a stopper and 6 mL ice-cold buffer G supplemented with 5 µg/mL α-chymotrypsin were added to perform on-column digestion overnight at 4 °C in the fridge. Protease digestion was stopped the next morning by the addition of PMSF to 1 mM and incubation for 30 min on ice. The resulting eluate was then collected, and the column eluted once more with 12 mL buffer G. The pooled eluates were then concentrated over a 10 kDa cutoff spin column (Thermo Fisher Scientific, Karlsruhe, Germany) at 6000× *g* for 10 min at 4 °C. The concentrate volume was adjusted to 900 µL by the addition of buffer G and actin polymerization was induced by adding 100 µL of 10× MEK (20 mM MgCl2, 50 mM glycol-bis(2-aminoethylether)-N,N,N′,N′-tetraacetic acid (EGTA) and 1 M KCl). After 1 h polymerization at room temperature (RT), the F actin was sedimented by centrifugation at 31,500× *g* for 90 min at RT in a Hettich Mikro 22R benchtop centrifuge. The resulting F-actin pellet was resuspended in 1xMEK in buffer G and kept for up to several weeks at 4 °C or aliquoted and frozen at −80 °C for long-term storage.

### 4.8. SDS-PAGE and Western Blot Analysis

Fusion proteins were analyzed by SDS-PAGE standard methods using 9% polyacrylamide ready-to-use gels (anamed Elektrophorese GmbH, Groß-Bieberau, Germany). Gels were either stained with coomassie blue staining reagent or transferred to PVDF membranes using an electrophoresis transfer system (Bio-Rad Laboratories GmbH, Feldkirchen, Germany). Imp-PM19 and Imp-PM19 mutant fusion proteins were detected using monoclonal anti-His antibodies (Quiagen, Hilden, Germany) specific for the His-tag, following anti-Mouse-POD. Bound antibodies were detected by an enhanced chemiluminescence reaction (Thermo Fisher Scientific, Karlsruhe, Germany).

### 4.9. Labelling F-Actin with Phalloidin Conjugated with Alexafluor 555

Rabbit muscle or recombinant apple G-actin was diluted at 0.05 µg/µL in a general actin buffer. The F-actins were labelled with Acti-stain™ fluorescent phalloidin 555 (Cytoskeleton, Inc., Denver, CO, USA) during F-actin polymerization as recommended by manufacture. The labelled F-actins were centrifuged at 31.500× *g* for 90 min. The F-actin pellets were dissolved in actin buffer and visualized under a confocal microscope with a bandpass 575–615 nm filter.

### 4.10. Co-Sediment Assay

Rabbit muscle (Cytoskeleton, Inc., Denver, CO, USA) G actin (actin_rabbit_) was diluted at 0.05 µg/µL in a general actin buffer (G buffer, 5 mM Tris-HCl, pH 8.0 and 0.2 mM CaCl2). Polymerization (F-actin) was induced for 1 h at RT in F-buffer by adding of actin polymerization inducer (50 mM KCl, 2 mM MgCl2, 1 mM ATP) in the G buffer. Purified recombinant Imp-PM19-His and mutant were added to F-actin_rabbit_ and the mixture was further incubated at 4 °C for 1 h. Proteins were pelleted by centrifugation at 100,000× *g* for 30 min. The pellets were resuspended with 4× protein loading dye and analyzed by an SDS-PAGE followed by Western blot analysis.

### 4.11. Localization of Imp-PM19 and Imp-PM19 Mutant in Planta

The genes encoding full-length *Imp-PM19* and *ΔD1-∆D3 Imp-PM19* mutants were fused upstream to a GFP gene by subcloning them in the binary vector pPZP200:GFP [32] resulting in pPZP200:*Imp-PM19-GFP* and pPZP200:*ΔD1/ΔD2* or *∆D3-Imp-PM19-mutant-GFP*. In order to co-express Imp-PM19 with an actin-binding Talin protein of mouse (kindly provided by Dr. Elison Blancaflor, The Samuel Roberts Noble Foundation), the whole expression cassette of the kanamycin resistance gene (NPTII) at the left border of the pPZP200:*Imp-PM19-GFP* and its mutants plasmids was substituted for the expression cassette of *mTalin-DsRED2* (35S:Promoter-mTalin-DsRED2:Nos-terminator), as described by Boonrod and co-workers [13], resulting in pPZP200:*Imp-PM19-GFP/mTalin-DsRED2* and pPZP200:Δ*D1/*Δ*D2* or ∆*D3-Imp-PM19- mutant-GFP/mTalin-DsRED2*. The plasmids were transformed into *Agrobacterium tumefaciens* (*A. tumefaciens*). Agroinfiltration was performed as previously described [33]. The agrobacterial suspension carrying either pPZP200:*Imp-PM19-GFP/mTalin-DsRED2* or pPZP200:Δ*D1/ΔD2 or* ∆*D3-Imp-PM19 mutant-GFP/mTalin-DsRED2* was infiltrated into leaves of *N. benthamiana* plants using a 1 mL syringe. The infiltrated leaves were collected 2 days post Agrobacterium-mediated infiltration (pif). The expression and localization of the proteins were examined using a Zeiss Observer Z1 with LSM510 confocal laser scanning head (Carl-Zeiss AG, Oberkochen, Germany).

### 4.12. Single-Molecule Force Spectroscopy Using Scanning Force Microscopy

#### SFM Cantilever-Tip Functionalization

Silicon cantilevers (MSNL-10, Tip D) with a nominal spring constant of 0.03–0.06 N/m were purchased from Bruker, USA. Before use, the cantilevers were first cleaned in chloroform (Uvasol, chemicals dispensary/supply Department of Chemistry, TU Kaiserslautern) three times for 10 min, followed by treatment with oxygen plasma (50 W power, 10–20 sccm (standard cubic centimeters per minute) O2, 0.3–0.6 mbar pressure, 5 min; Pico, Diener electronics GmbH + Co. KG, Ebhausen, Germany). Next, the cleaned and oxidized cantilevers were transferred into an argon atmosphere for silanization via gas-phase evaporation as described by [33,34,35]. Thereafter the amino silane and annealed cantilevers were crosslinked by immersing them in the PEG-crosslinker solution as described by [36,37]. The PEG-crosslinker has two different functional ends: one end ensures the attachment of the linker to the aminated tip surface, while the other is reserved for the coupling of the probe molecule [38]. Due to its flexibility, the spacer PEG-crosslinker will allow the fixed ligand to move freely at low surface density, and the ligand can interact with its complementary ligand molecules. PEG-functionalized cantilevers were placed on Parafilm in a polystyrene Petri dish, and a drop of 200 µL of purified recombinant Imp-PM19 wildtype/Imp-PM19 mutant solution was pipetted onto the functionalized cantilevers. Next, a small amount of reducing agent, sodium cyanoborohydride (NaCNBH_3_, Fluka Chemie GmbH, Buchs, Switzerland), was given to the Imp-PM19/∆D3-Imp-PM19 mutant (drop) solution for 1 h and mixed carefully to stabilize the covalent binding of ligand to the spacer. After 1 h of incubation, 10 µL of ethanolamine solution was added to the drop mixture for 10 min. Finally, the Imp-PM19/∆D3-Imp-PM19 mutant cantilevers were washed three times for 5 min in PBS buffer solution (pH 7.3).

### 4.13. Receptor Preparation and Immobilization on the Mica Substrate

Freshly cleaved muscovite mica substrate is ideal for SMFS studies even though its surface is negatively charged in a neutral pH value [18]. Receptors (protein) should be tightly immobilized to the mica substrate because loose receptor attachment would lead to a pull-off of the receptor from the surface by the ligand on the cantilever tip. This pull-off effect would block ligand-receptor recognition and obscure the recognition force experiments [18]. Since mica and F-actin proteins are negatively charged in the imaging buffer solution (F-Buffer, pH 6.8), the mica surface must be firstly modified to allow the adsorption of F-actin proteins. The ultra-flat mica substrate was first cleaned with adhesive tape and secondly in an O_2_-plasma (50 W power, 10–20 sccm (standard cubic centimeters per minute) O_2_, 0.3–0.6 mbar pressure, 5 min; Pico, Diener electronics GmbH + Co. KG, Ebhausen, Germany). After this cleaning, the negatively charged mica surface was positively modified by incubating in 0.01% poly-l-lysine (m.w. 30,000–70,000 g/mol, Sigma-Aldrich, St. Louis, MO, USA) for 15 min. The 0.01% poly-l-lysine solution was prepared in ultrapure water. After that, the poly-l-lysine mica was rinsed with distilled water and dried under nitrogen flow. Next, the dried and positively charged mica surface was immersed in F-actin_rabbit_ protein solution with unlabeled phalloidin for 15 min and subsequently rinsed with F-actin buffer (F- buffer).

To avoid dying of the immobilized F-actin_rabbit_ filaments on mica plates, the plates are incubated in the imaging buffer for further experiments.

### 4.14. Measuring Interaction Forces between Imp-PM19/∆D3-Imp-PM19 and Actin Proteins

Single-molecule force spectroscopy (SMFS) measurements were performed using a Nano wizard3 Bio (JPK Instruments, Berlin, Germany) scanning force microscope (SFM). All the measurements were measured in a liquid environment, using imaging buffer (F-buffer, pH 6.8) to allow both proteins to be in their native conformation. In addition, using a PFTE ring on the sample top reduces the evaporation of the solution during the measurements.

Before measuring the force curves, the cantilever’s sensitivity and force constant were determined using the contact-based method (included in the JPK Nano wizard 3 software) [39,40]. Afterwards, the single-molecule force spectroscopy was measured using a force setpoint of 800 pN. After recording the single-molecule force curves, the retract curves were analyzed with data processing software using a worm-like chain (WLC) as a standard model for the fit [41]. Unbinding events were analyzed beginning with 10 nm retraction length because of the length of the PEG crosslinker (less than 8 nm, approximately ~6 nm). However, the dimension of the attached and folded ligand protein is unknown. Therefore, the height of 4 nm was chosen as a supposed (minimal) height of Imp PM19 or ∆D3-Imp-PM19 mutant.

### 4.15. Statistical Analysis

The normal distribution of the data was checked in the course of the evaluation of the SFM data in the histogram using a normal distribution curve. A statistical *t*-test was performed to determine the significance of the effect of both ligands on the actin filament (adhesion forces and work of adhesion). Since it is assumed that the Imp PM19 has an actin-binding region and ∆D3-Imp-PM19 mutant does not, the one-side test is used to underline this difference. The null hypothesis (H_0_) says there is no difference between the interaction of both ligands on the F-actin protein. This hypothesis can be rejected or retained. The alternative hypothesis is accepted when the null hypothesis is rejected, and vice versa. *P*-value, t-statistic, and critical *t*-value were calculated for the evaluation. *P* ≤ 0.05 (alpha level) was considered statistically significant. Th one side test is used to compare the t-statistic to the critical *t*-value. P (T <= t) one-sided is smaller than the alpha level (0.05), i.e., a significant effect is observed. Therefore, the Imp-PM19 protein and ∆D3-Imp-PM19 mutant do not have the same effect on the F-actin protein. Then, the null hypothesis is rejected, and the alternative hypothesis (that the effect of both ligands on the F-actin protein is not equal) is accepted. In addition, the t-statistic is larger than the critical *t*-value in the one side *t*-test, which is relevant for us to show this assumption (effect), and the null hypothesis is rejected.

## Figures and Tables

**Figure 1 ijms-24-00968-f001:**
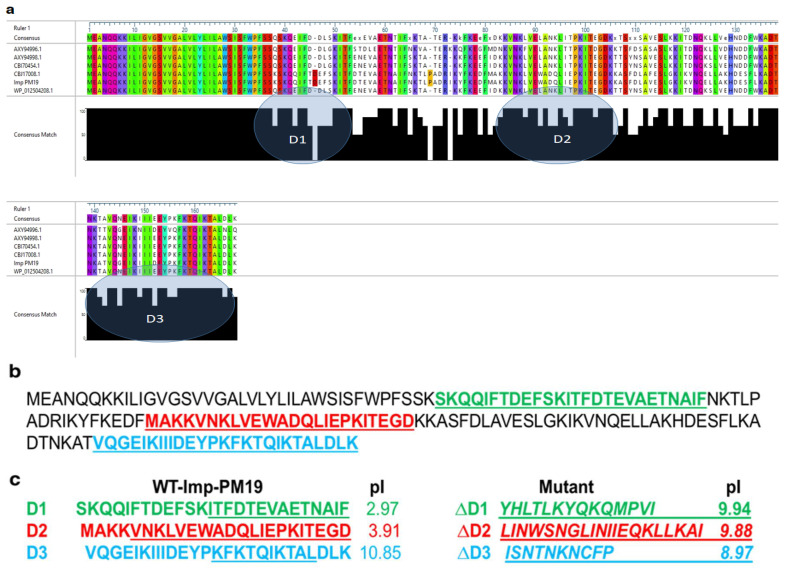
Amino acid sequence alignment of Imp-PM19 of different ‘*Ca*. P. mali’strains. (**a**) Amino acid sequences of Imp obtained from Gene Bank were aligned using DNA star program. Color letter indicate conserved amino acids. The grey shaded area (circled) in consensus math indicates the conserved regions. (**b**) Protein sequence of Imp-PM19 and (**c**) mutants. Green, Red and Blue labeled are D1-D3 conserved regions and their pI which were mutated (underlined letters) to generate ΔD1-∆D3*-Imp-PM19*-mutants (italic and underlined letters), respectively.

**Figure 2 ijms-24-00968-f002:**
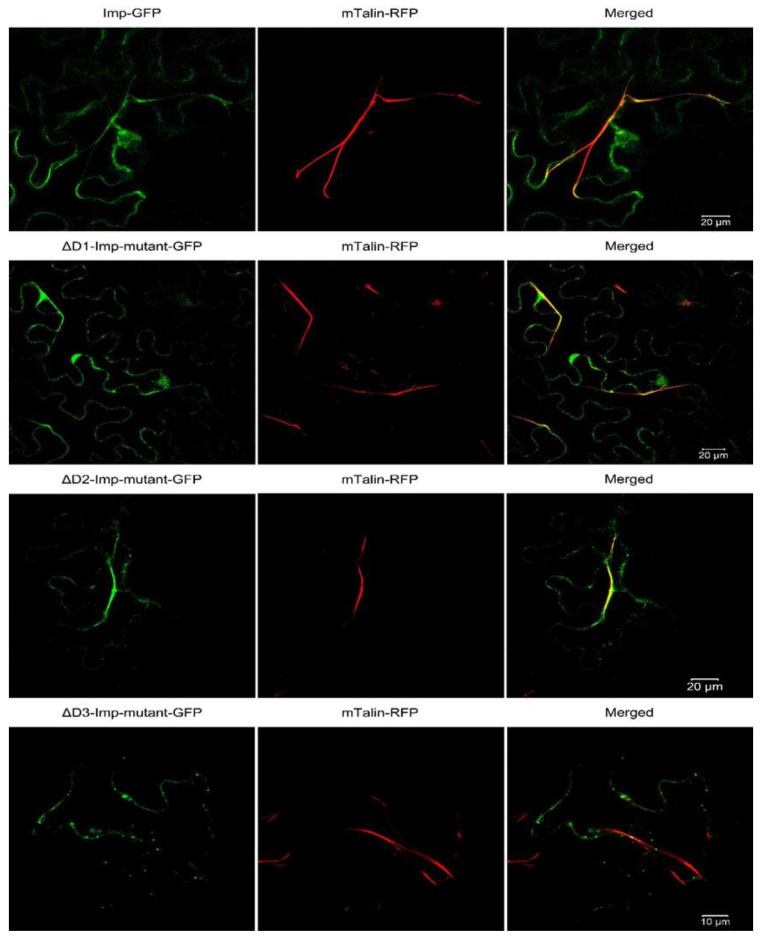
Co-expression of Imp-PM19, its mutants and mTalin in planta. Imp-PM19 and its mutants were fused with GFP and co-transiently expressed with mTalin fused to RFP to mark the plant actin. The localization of expressed proteins were analyzed by visualizing the infiltrated leaves mesophyll under confocal microscopy using GFP and RFP filters. The F-actin can be clearly seen as a long stretch filament (Red colour) in the middle panel. Co-localization of Imp-GFP and mTalin-Red is indicated by yellow colouring in the merger. A magnification of Figure 2 can be seen in Appendix A.

**Figure 3 ijms-24-00968-f003:**
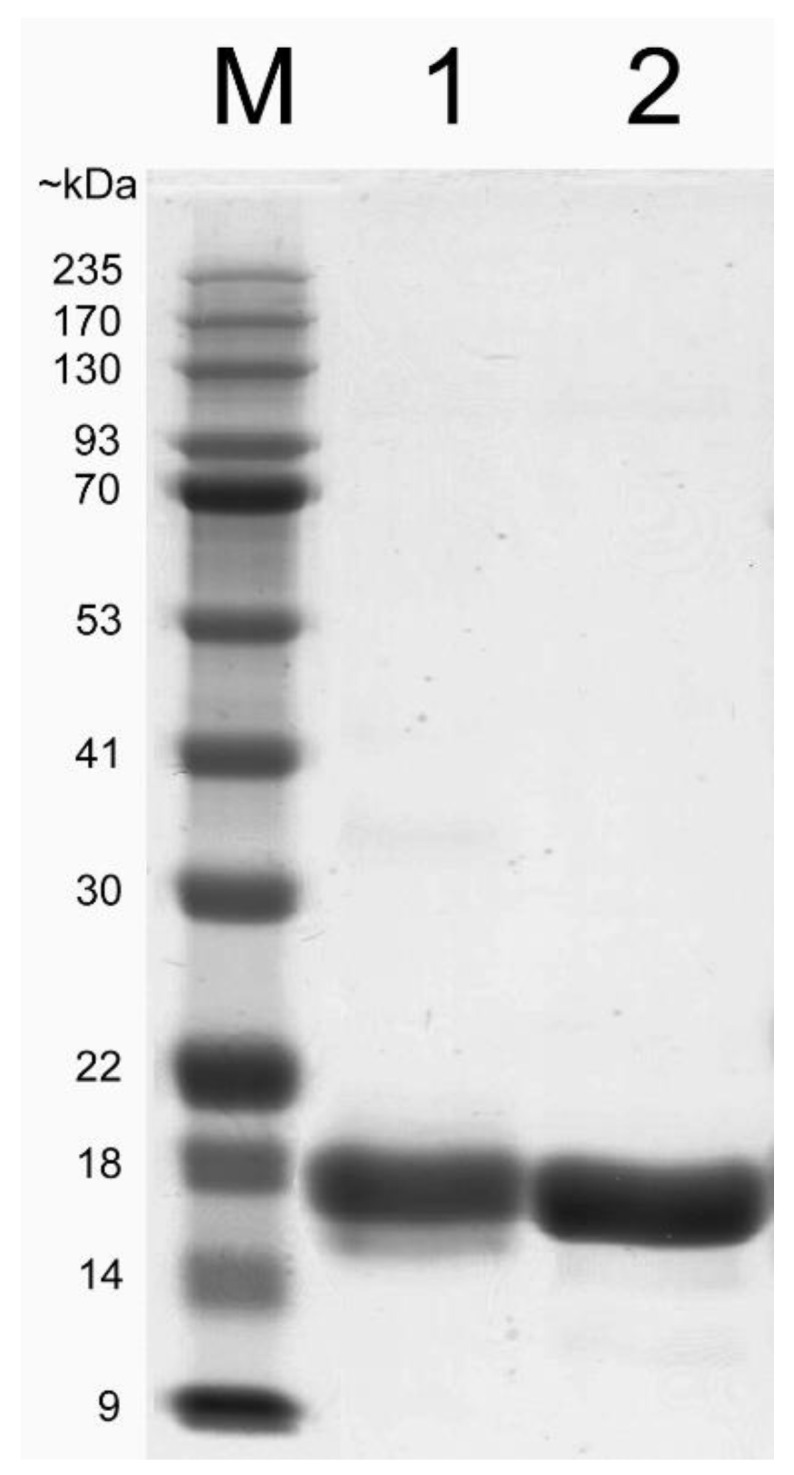
Coomassie blue straining SDS-PAGE of purified recombinant Imp-PM19 and ∆D3-Imp-PM19 mutant. His-tagged *Imp-PM19* and *∆D3-Imp-PM19*-mutant were expressed recombinantly in *E. coli*. The proteins were purified using Ni-NTA fast flow column. M is a protein marker. Lane 1 and 2 are purified Imp-PM19-His and ∆D3-Imp-PM19-His, respectively.

**Figure 4 ijms-24-00968-f004:**
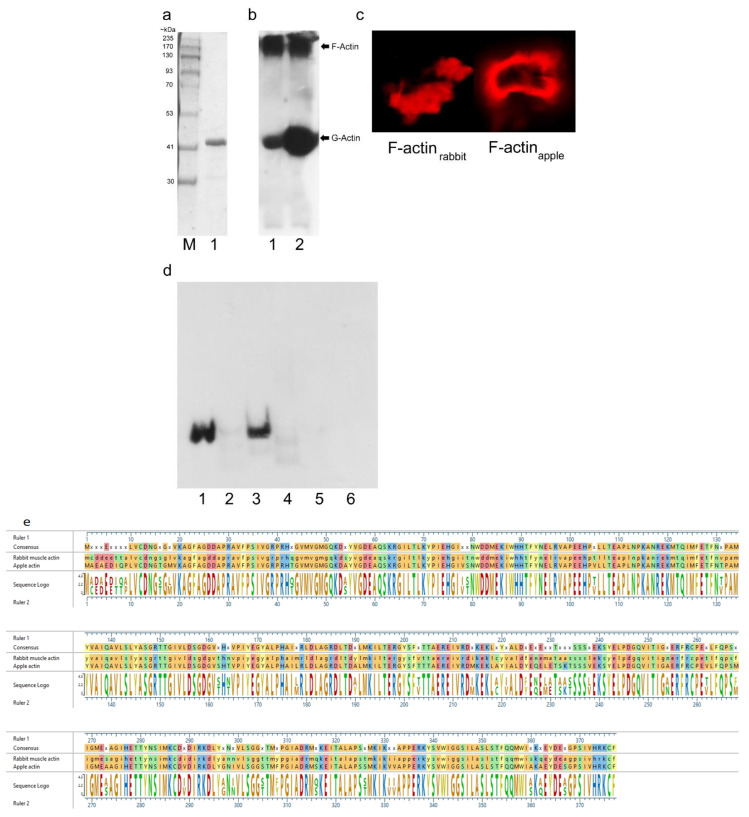
Co-sedimentation assay with recombinant actin_apple_, Imp-PM19 and ∆D3-Imp-PM19-mutant. (**a**) Coomassie-stained SDS-PAGE of recombinant apple actin. M is a protein marker. 1 is recombinant apple actin. (**b**) Western blot of recombinant actin_apple_. Recombinant actin_apple_ (lane 1) and actin_rabbit_ (lane 2, a positive control) were polymerized and subjected to SDS-PAGE. After blotting, the proteins were detected using anti-actin and anti-mouse-POD, respectively. (**c**) Labelling F-actins with Acti-stain™ fluorescent phalloidin 555 (Cytoskeleton, Inc). The labelled F-actins were visualized under a confocal microscope with a bandpass 575–615 nm filter. (**d**) Co-sedimentation assay. Imp-PM19 and ∆D3-Imp-PM19 mutant were incubated with F-actin_apple_ and F-actin_rabbit_. After height speed centrifugation, the protein pellets were subjected to SDS-PAGE, and a western blot was performed using anti-His and anti-mouse-POD, respectively. Lane 1 and 2 are co-sedimented Imp-PM19 (1) and ∆D3-Imp-PM19-mutant (2) with F-actin_rabbit_. Lane 3 and 4 are co-sedimented Imp-PM19 (3) and ∆D3-Imp-PM19-mutant (4) with recombinant F-actin_apple_. Lane 5 and 6 are BSA (negative controls) co-sedimented with Imp-PM19 (5) and ∆D3-Imp-PM19-mutant (6). (**e**) An alignment of rabbit muscle actin (gene ID: 10152413) and apple actin (gene ID: XP_008355144.1). The alignment result shows 88% identity of the two actins.

**Figure 5 ijms-24-00968-f005:**
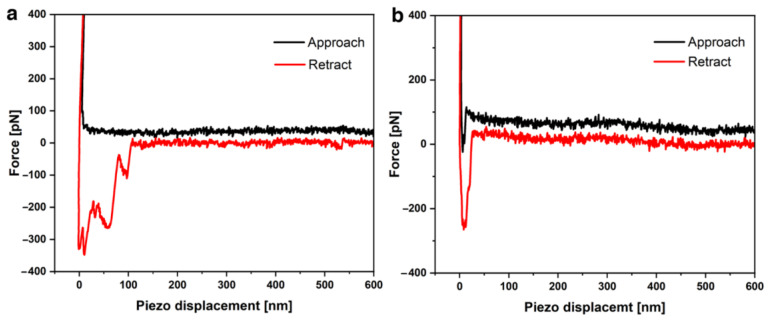
Single molecule force distance curves of an Imp-PM19/∆D3-Imp-PM19 mutant functionalized SFM-tip on the F-actin_rabbit_ surface measured by SFM. The interactions of F-actin_rabbit_ with the Imp-PM19 (**a**) and the ∆D3-Imp-PM19 mutant (**b**) are represented.

**Figure 6 ijms-24-00968-f006:**
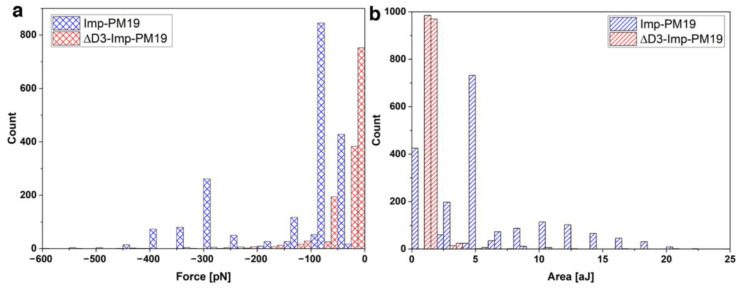
Comparing the obtained interaction parameters with Imp-PM19 (blue bar) and the ∆D3-Imp-PM19 mutant (red bar) on F-actin_rabbit_ filament. The work of adhesion (**a**) and the most frequently onserved adhesion (**b**) of Imp-PM19 and ∆D3-Imp-PM19 with F-actin_rabbit_ filament ligands are measured by SMFS. The results are graphically represented. Retraction speed 2 µm/s, number of curves = 1000, set point of 800 pN.

**Figure 7 ijms-24-00968-f007:**
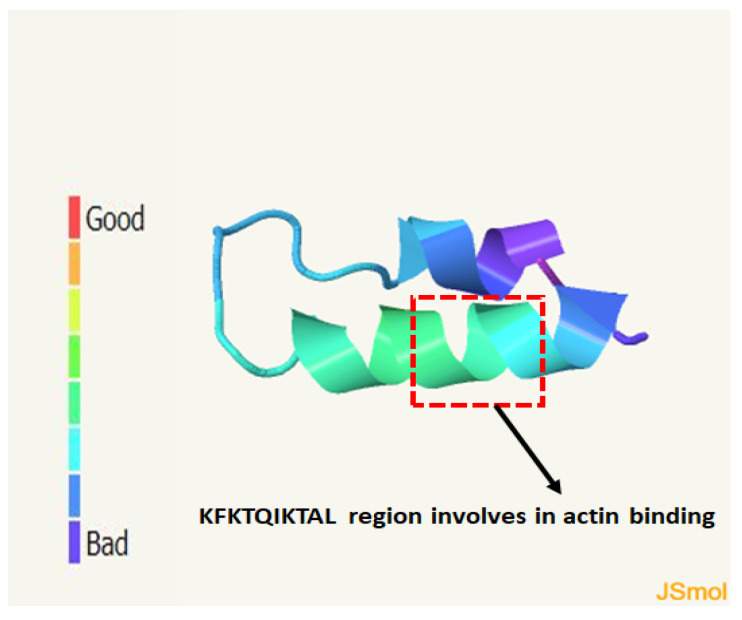
Prediction of Imp protein folding using Phyre2 computing program. The protein sequence of Imp-PM19 was entered to Phyre2 computing program. The result shows that Imp-PM19 folds as alpha-alpha-superhelix form with confidence 17.72%. The colour scale indicates confidence score.

## Data Availability

All data generated or analyzed during this study are included in this published article.

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
