# Peer review of "Identification of the Actin-Binding Region and Binding to Host Plant Apple Actin of Immunodominant Transmembrane Protein of ‘*Candidatus* Phytoplasma mali’"

_ijms, 2023, doi:10.3390/ijms24020968_

Round 1
Reviewer 1 Report
In this work, K. Boonrod and co-authors identify an acting-binding region in the Ca. P. mali Imp-PM19 protein by means of interdisciplinary approaches. This manuscript can be relevant to a reasonable number of scientists working in the field of phytoplasma-host interactions. This reviewer thinks that the presented work is overall solid; however a list of points (described below) should be addressed for the acceptance of the manuscript.
1. In the Introduction authors should add a paragraph detailing the phytopathological features of Ca. P. mali and its impact in agriculture. This will help readers not expert in the field.
2. Figure 1: the content of panel A is not visible at all. I recommend to either enlarge this panel or to dedicate a Figure only to this alignment. I also recommend to include in the Figure (or in the main text) a description about the specific actin sequence(s) that are recognized by Imp-PM19.
3. Figure 2: the figure legend lacks of a clear description of the plant ultrastructures visible in the micrographs. Then, in the Merged panels, I suggest to include a higher (eventually digital) magnification to better highlight the co-localization signals of Imp-PM19 with actin.
4. Paragraph 2.5, lines 151-153. The choice of using rabbit actin instead of apple actin is not fully convincing. Authors should include a statement explaining, for instance, that the two actin sequences recognized by Imp-PM19 are highly conserved and eventually show an alignment.
5. Discussion. Unless the structure of Imp-PM19 has been already solved (which is not clear from the information present in the manuscript), in the era of protein structure prediction I believe that this paper will be more informative if the authors will show a 3D model of Imp-PM19 protein generated by protein folding recognition servers, such as AlphaFold or Phyre2. In this predicted model authors should highlight the domain involved in the binding with actin. Additionally, I would expect to see a more elaborated hypothesis that explains how phytoplasms invade the plant cell by means of actin binding and how this information is useful to combat these phytopathogens.
Author Response
Dear Review,
Thank you for your evaluation of our MS and your valuable suggestion. We responded to your comments as follows;
- In the Introduction authors should add a paragraph detailing the phytopathological features of Ca. P. mali and its impact in agriculture. This will help readers not expert in the field.
Added
- Figure 1: the content of panel A is not visible at all. I recommend to either enlarge this panel or to dedicate a Figure only to this alignment. I also recommend to include in the Figure (or in the main text) a description about the specific actin sequence(s) that are recognized by Imp-PM19.
Done
- Figure 2: the figure legend lacks of a clear description of the plant ultrastructures visible in the micrographs. Then, in the Merged panels, I suggest to include a higher (eventually digital) magnification to better highlight the co-localization signals of Imp-PM19 with actin.
A clear description of plant actin filament was added in the figure legend. A digital figure was included in Supplement.
- Paragraph 2.5, lines 151-153. The choice of using rabbit actin instead of apple actin is not fully convincing. Authors should include a statement explaining, for instance, that the two actin sequences recognized by Imp-PM19 are highly conserved and eventually show an alignment.
Added
- Discussion. Unless the structure of Imp-PM19 has been already solved (which is not clear from the information present in the manuscript), in the era of protein structure prediction I believe that this paper will be more informative if the authors will show a 3D model of Imp-PM19 protein generated by protein folding recognition servers, such as AlphaFold or Phyre2. In this predicted model authors should highlight the domain involved in the binding with actin. Additionally, I would expect to see a more elaborated hypothesis that explains how phytoplasms invade the plant cell by means of actin binding and how this information is useful to combat these phytopathogens.
Added
Reviewer 2 Report
Phytoplasma diseases of apple trees are regarded as one of the major impediments to production in many parts of the world, resulting in significant yield and quality losses. Frankly, the article entitled "Identification of the actin-binding region and binding to host plant apple actin of Immunodominant transmembrane protein of ‘Candidatus Phytoplasma mali’" is, in general, well written, great of interest, and will contribute to a better understanding of phytoplasma-host interactions and serves in phytoplasma disease control programs. It contains a good introduction to the experiment. Moreover, I think the authors have sufficiently related their results to those previously published in the world literature. Overall, I believe that the article needs minor linguistic corrections and shows some petty points. However, I think that the conclusion section should be rewritten, taking into consideration the comments shown in the revised manuscript attached.

Author Response
Dear Editor,
Thank you very much for your evaluation and suggestion. We responded to your suggestion and comments as folllow:
All points were corrected as recommend.
Comment1: The ref [3] was replaced as recommendation.
Comment 2: regarding to point 2.6. It was removed to M&M as suggestion.
Comment 3: Conclusion. We improved our conclusion as recommendation.
Comment4; in M&M 4.14. Normal distribution can be checked either with the tests mentioned by the reviewer or with a histogram. The SFM data evaluation is based on a histogram associated with normality check. Thus, the data were checked for normal distribution during the SFM data evaluation.
Round 2
Reviewer 1 Report
Authors have correctly replied to all my recommendations and the manuscript has been sufficiently improved to warrant publication in IJMS.